# Doolittle: Benchmarks and Corpora for Academic Writing Formalization

**Shizhe Diao**[1*], **Yongyu Lei**[1*], **Liangming Pan**[2], **Tianqing Fang**[1],
**Wangchunshu Zhou**[3], **Sedrick Scott Keh**[4], **Min-Yen Kan**[5], **Tong Zhang**[1]

[1]The Hong Kong University of Science and Technology
[2]University of California, Santa Barbara [3]ETH Zürich [4]Toyota Research Institute
[5]National University of Singapore
{sdiaoaa, yleiah, tfangaa}@connect.ust.hk
tongzhang@ust.hk

## Abstract

Improving the quality of academic writing is a meaningful but challenging task. Conventional methods of language refinement focus on narrow, specific linguistic features within isolated sentences, such as grammatical errors and improper word use. We propose a more general task, *Academic Writing Formalization (AWF)*, to improve the overall quality of formal academic writing at the paragraph level. We formulate this language refinement task as a formal text style transfer task which transfers informal-academic text to formal-academic and contribute a large-scale non-parallel dataset, Doolittle, for this purpose. Concurrently, we apply a method named metric-oriented reinforcement learning (MORL) to two pretrained language models (PLM) where we incorporate different levels of automatic feedback into the training process. Our experiments reveal that existing text transfer models and grammatical error correction models address certain aspects of AWF but still have a significant performance gap compared to human performance. Meanwhile, language models fine-tuned with our MORL method exhibit considerably improved performance, rivaling the latest chatbot Chat-GPT, but still have a non-negligible gap compared to the ground truth formal-academic texts in Doolittle.[1]

## 1 Introduction

Writing in a second language often leads to characteristic errors. In English, such errors include subject–verb disagreement, noun–number disagreement, and determiner misuse (Lado, 1957; Rod, 1994). Therefore, a language refinement system with the ability to suggest or automatically correct such errors is highly desirable (Yuan and Felice, 2013; Rozovskaya and Roth, 2016). Towards this end, research in grammatical error correction

| |
|---|
| [S]: We propose more sophisticated hierarchical model to include geographical *informations*. |
| [T]: We propose *a* more sophisticated hierarchical model to include geographical *information*. |
| [S]: This is because the teaching and learning on *science* domain relies *much* on the ability of reasoning and computation, which directly utilizes the *advantage of computer*. |
| [T]: This is because the teaching and learning on *a scientific* domain relies *considerably* on the ability of reasoning and computation, which directly utilizes the *advantages of computers*. |
| [S]: METEOR is another n-gram overlap measure initially designed for evaluating machine translation systems. ROUGE-L is a commonly-adopted metric for text summarization. |
| [T]: Both METEOR and ROUGE-L specialize in BLEU's n-gram overlap idea for machine translation and text summarization evaluation, respectively. |

Table 1: Informal-academic paragraphs with formal-academic rewrites, denoted S and T, respectively. The refined form is highlighted blue, the original in red.

(GEC) (Ng et al., 2014; Bryant et al., 2019) focuses on identifying and correcting many of such grammatical errors. However, even if non-native speakers can write grammatically correct sentences, their language use is sometimes less concise and fluent than those written by native speakers (Lado, 1957; Rod, 1994). For example, the two source sentences in the third example shown in Table 1 are grammatically correct but less fluent due to their structural redundancy. They sound more fluent when combined into a single sentence by adding an appropriate conjunction.

In light of this, we propose the novel task of *Academic Writing Formalization (AWF)* that aims to generalize the scope of GEC for language refinement: given an informal-academic paragraph $\mathcal{P}$, the objective of AWF is to refine the language of

---

*Equal Contribution.
[1]The datasets and code are available at https://github.com/shizhediao/Doolittle.

$\mathcal{P}$ to make it grammatically correct, concise, and fluent, while preserving its semantics. Different from GEC, which solely concentrates on grammatical error correction for a single sentence, AWF works on paragraph-level contexts and aims for refinements beyond grammar. This requires the model to comprehend and then rephrase the entire paragraph.

Specifically, AWF considers three objectives in academic writing formalization to refine the language. 1) *grammar correction*: correcting grammatical errors in the paragraph, the objective of GEC. 2) *word refinement*: replacing inaccurate words and phrases with more accurate and concise ones. Evidence has shown that there are vital differences in vocabulary usage between native and non-native writers (Malmasi et al., 2017). For example, we replace "science" and "much" with "scientific" and "considerable" in Table 1's second example. 3) *structure modification*: changing the sentence or paragraph structure to convey the meaning more concisely. For example, the third example in Table 1 combines two similar sentences into a single long sentence to convey the information more efficiently.

Although there exist several large-scale corpora for GEC (Yannakoudakis et al., 2011; Tajiri et al., 2012; Mizumoto et al., 2012; Dahlmeier et al., 2013; Napoles et al., 2017; Bryant et al., 2019), they either focus on word/phrase/sentence level text refinement or do not target scientific texts, which makes none of them available for AWF. We thus construct DOOLITTLE[2], a large-scale, non-parallel dataset containing 55.6K formal-academic paragraphs and 13.0K informal-academic ones. DOOLITTLE is based on the Semantic Scholar Open Research Corpus (S2ORC; Lo et al., 2020). For each paragraph, we ask human annotators to rate the academic formality of the language via crowdsourcing. Expert annotators then refine the language for around 900 paragraphs to obtain a parallel corpus that serves as the development and test sets of DOOLITTLE.

To investigate the performance of state-of-the-art models for AWF, we adopt four baseline models of text style transfer, two baseline models from low-resource GEC, the widely-available large language model (LLM) ChatGPT, and two pretrained language models (PLM) fine-tuned with our pro-

posed method: metric-oriented reinforcement learning (MORL). We find that style transfer models are unsuccessful in discriminating the differences between formal and informal text, resulting in lower scores for academic formality, perplexity, and meaning preservation, while GEC baselines perform relatively better across all metrics but only marginally modify the inputs. On the other hand, BARTLarge (Lewis et al., 2020) and Galactica-1.3B (Taylor et al., 2022) fine-tuned with our MORL approach provide comparable GEC results to ChatGPT, despite having significantly fewer parameters. Nonetheless, none could comprehensively outperform the ground truth formal-academic paragraphs. It is worth noting that metric-oriented RL has been explored in the context of text generation, with some early studies (Wu et al., 2016; Choshen et al., 2020) using RL for neural machine translation optimized by BLEU, but with limited success. To the best of our knowledge, we are the first to demonstrate that applying metric-oriented RL to PLMs yields promising results, indicating that metric-based RL is well-suited for powerful backbone models.

We summarize our contributions as follows: 1) We propose a novel setting for paragraph-level language refinement, formulating it as a text style transfer problem. 2) We construct DOOLITTLE, the first large-scale dataset for academic writing formalization. Considering that AWF is a common use case of LLMs such as ChatGPT, we believe DOOLITTLE can serve as a good testbed for benchmarking LLMs. 3) We propose a method, metric-oriented reinforcement learning (MORL), and show its effectiveness and cost-efficiency in tuning PLMs. 4) We conduct a comprehensive evaluation of neural approaches on our task and show that their performance still suffers from a sizable gap compared to formal-academic rewrites by humans. This highlights the need for the dataset and the AWF task.

## 2 Related Work

**Language Refinement.** There are two tasks typical of language refinement, both focusing on enhancing the quality of sentences. *Post-editing* (Novak et al., 2016; Xia et al., 2017; Guu et al., 2018; Freitag et al., 2019) is designed to rectify typical errors in machine translation, thereby augmenting the generation quality, as measured by BLEU. The other task, *Grammatical Error Correction*

---

[2]Named after the lower-class protagonist from the English film *My Fair Lady*, who undergoes training to transform her accent and manners into one of a proper lady.

*(GEC)*, is formulated as a parallel translation task with phrase-based machine translation (PBMT) models (Rozovskaya and Roth, 2016; Junczys-Dowmunt and Grundkiewicz, 2016), neural machine translation (NMT) models (Chollampatt and Ng, 2018; Junczys-Dowmunt et al., 2018), and hybrid PBMT–NMT models (Grundkiewicz and Junczys-Dowmunt, 2018). However, these methods require large amounts of parallel data which are expensive to annotate. To address this, low-resource GEC (Bryant et al., 2019) builds models that do not rely on large parallel data. Choe et al. (2019), Grundkiewicz et al. (2019), and Zhou et al. (2020) initially pretrain a transformer model with large synthetic parallel corpora which are generated from a realistic noising function and then fine-tune this model on a small in-domain parallel dataset.

**Reinforcement Learning from Human Feedback (RLHF).** As a notable advancement within the realm of reinforcement learning (RL), reinforcement learning from human feedback (RLHF) integrates human feedback into the training process. This approach trains a model to align more closely with user intentions, thereby equipping the model with the ability to generate more reliable, authentic, and useful results (Ziegler et al., 2019; Ouyang et al., 2022; Dong et al., 2023). RLHF manifests its advancement and convincing capability in the recent state-of-the-art chatbot ChatGPT (OpenAI, 2022). The fundamental workflow of RLHF can be succinctly summarized in three steps below: **Step 1:** Train a policy model with supervised training on collected demonstration data. **Step 2:** Train a reward model on collected comparison data. **Step 3:** Optimize the policy against the reward model using reinforcement learning with proximal policy optimization (PPO) algorithm (Schulman et al., 2017).

## 3 Dataset Construction

We present DOOLITTLE, a corpus of academic formality non-parallel texts from scientific paper sources (§ 3.1), where we manually annotate academic formality by crowdsourcing to obtain a large set of non-parallel training paragraphs in two styles (§ 3.2). We then conduct a second annotation task called formal-academic rewrite in order to create a small parallel dataset for evaluation (§ 3.3).

### 3.1 Data Source

Our focus lies on scientific texts, encompassing both published articles and preprints as our primary sources. These scientific articles are typically of high quality, usually having been proofread, allowing models to focus on improvements in terms of lexical choice and sentence structure. We use the Semantic Scholar Open Research Corpus (S2ORC) (Lo et al., 2020), a large corpus of 81.1 million English scientific papers spanning many academic disciplines including medicine, biology, computer science, and so on. There are four reasons for choosing S2ORC: 1) It is a clean dataset of scientific papers, which are of good quality without trivial mistakes; 2) It exposes rich metadata, inclusive of paper titles, authors, published venue, and year of publication; 3) It provides full text for 81.1 million open access papers without copyright issues, so that both the distribution of the data and replication of our work are possible without infringement; 4) The full text preserves meaningful structures such as paragraph breaks and section headers, so that the text is easily extracted.

### 3.2 Academic Formality Annotation

We now describe how we set up the annotation crowdsourcing tasks to mark the academic formality of each paragraph. We first randomly sample a subset of 90,000 short paragraphs which composed of more than 2 sentences with lengths between 20 to 100 words. Then we classify them into formal-academic and informal-academic paragraphs. We adopt an unbiased means that ignores the paper authors and instead asks human annotators to rate the academic formality of each paragraph via crowdsourcing.

**Annotation Task Overview.** Each annotation task contains 100 paragraphs. Annotators are asked to score each paragraph from 1 (sounds informal-academic) to 5 (sounds formal-academic). For any paragraph that contains incomplete sentences, an assignment of 0 is acceptable. We provide a detailed annotation guideline to illustrate the standards for different scores. For example, a part of the standard for rating a score of 2 is as follows: "The language expression is unclear that you cannot fully understand the meaning... " We had four experts to construct a quality control test consisting of 500 examples, which we randomly inject into each task. The detailed descriptions for each score with corresponding examples and gold set construction

| | | P# | S# | V# | Avg. Words | Avg. Sent. | ACC-cola | ACC-aesw | PPL | SIM | ED | BARTS |
|---|---|---|---|---|---|---|---|---|---|---|---|---|
| Train | FA | 55.6K | 172.8K | 84.3K | 51.42 | 3.11 | 97.56 | 79.64 | 24.44 | - | - | |
| | IFA | 13.0K | 41.3K | 38.9K | 52.17 | 3.17 | 95.81 | 68.51 | 32.56 | - | - | |
| Dev | FA | 465 | 1359 | 5.2K | 47.33 | 2.92 | 98.49 | 78.27 | 31.19 | 98.75 | 11.03 | -1.19 |
| | IFA | 465 | 1362 | 5.3K | 47.79 | 2.92 | 95.69 | 72.04 | 33.07 | | | |
| Test | FA | 415 | 927 | 4.4K | 42.52 | 2.23 | 98.31 | 77.83 | 33.18 | 98.09 | 10.87 | -1.24 |
| | IFA | 415 | 910 | 4.5K | 43.08 | 2.19 | 95.66 | 69.64 | 35.97 | | | |

Table 2: The statistics of the DOOLITTLE dataset, where P#, S#, V#, Avg. Words, and Avg. Sents. refer to the number of paragraphs, number of sentences, vocabulary size, average words per paragraph, and average sentences per paragraph, respectively. We also report the transfer accuracy (ACC), perplexity (PPL), Semantic Similarity (SIM), char-level edit distance (ED), and BARTScore (BARTS). FA and IFA denote formal-academic and informal-academic, respectively.

are shown in the Appendix A.

**Publishing Annotation Tasks.** We conduct annotation on the Amazon Mechanical Turk (AMT) platform. For each annotation task, we randomly sample a paragraph from each of the five scores in the gold set for a total of 5 paragraphs with their corresponding gold scores. These are used as test cases for quality control and are amalgamated with 95 unannotated paragraphs. Each task is assigned to two annotators independently. Annotators are given 7 hours to complete the task. To refine the cohort of workers that are eligible to complete our task, we impose restrictions to include only annotators who are located in primarily English speaking countries, and who have finished at least 100 tasks before on the AMT platform with an approval rate above 90%.

**Quality Control.** We have the following standards to control the annotation quality:
• Time spent on each task should be greater than 500 seconds.
• Variance should be greater than a threshold $\epsilon_1$ to ensure not all scores are the same.

$$VAR = \frac{1}{n}\sum_{i=1}^{n}(x_i - \mu)^2 > \epsilon_1 \qquad (1)$$

• The allowed discrepancy with gold set annotations (defined below) must be smaller than a threshold $\epsilon_2$.

$$GAP = \sum_{i=1}^{5}|\text{annotation}_i - \text{gold}_i| < \epsilon_2 \qquad (2)$$

where $\text{gold}_i$ denotes the score of $i$-th test case and $\text{annotation}_i$ is its corresponding annotation. Annotations that can not meet all of the above standards are rejected, and we provide the workers with detailed reasons of rejection to help them improve

the quality of annotations. Any worker who repeatedly performs poorly (i.e., the rejection rate is above 50% and he/she has done over 6 tasks) will be eventually blocked from our tasks.

**Constructing DOOLITTLE.** We post-process the annotations into binary scores — 0 (informal-academic) or 1 (formal-academic) — using the following rules. Here, we define $S1$ and $S2$ as the scores given by Annotators 1 and 2, respectively.

- *Incomplete*: $S1 = 0$ or $S2 = 0$

- Informal-academic: $0 < S1 \leq \alpha$ and $0 < S2 \leq \alpha$

- Formal-academic: $S1 > \alpha$ and $S2 > \alpha$

- *Others*: the remaining data whose scores do not hold the above standards.

Paragraphs categorized under *Incomplete* and *Others* would be filtered out because we want a cleaner dataset with a high mutual agreement. $\alpha$ is set to 2 according to our definition of the scores, as indicated in the Appendix A.2. We thus obtain a large set of non-parallel paragraphs in two styles (Table 2): formal-academic and informal-academic. We randomly select 500 informal-academic paragraphs for development and another 500 for testing. The remainder is used as training set.

**Annotation Results.** We evaluate the disagreement between two annotators to check the quality of annotation by $Disagreement = |S1 - S2|$. We observed that 43.30% annotations have the same scores (disagreement is 0) and the disagreement of about half (56.70%) annotations is 1, which indicates there is a high rate of agreement. In addition, the Cohen's Kappa coefficient between two annotators is 0.657, showing a strong agreement between the two annotators' scoring.

### 3.3 Test Set Construction

Our methodology produces a large, non-parallel, binary-annotated corpus. To obtain a small parallel development and test set for evaluation, we then conduct formal-academic rewrites to produce a set of paragraph pairs. To ensure the quality of the development and test set, the two native speakers involved in the construction of the gold set, who are quite familiar with the standards of academic formality and possess a thorough understanding of our task, are asked to rewrite informal-academic paragraphs into formal-academic paragraphs. Subsequently, two authors of this paper reviewed the rewrites and rejected those that do not meet the required standards. The average time consumed for a rewriting task containing 100 paragraphs is 220 minutes. During this process, incomplete paragraphs were identified by annotators and removed from the final development and test set. The final statistics are shown in Table 2.

## 4 Dataset Analysis

### 4.1 Automatic Evaluation

• **Transfer Accuracy (ACC)** To capture the transfer success of academic formality in paragraph level, following Krishna et al. (2020), two RoBERTa-Large (Liu et al., 2019) models are fine-tuned on the CoLA corpus (Warstadt et al., 2019) and the automated evaluation of scientific writing shared task 2016 dataset (AESW) (Daudaravicius, 2015), serving as two academic formality classifiers respectively. The transfer accuracy on generated paragraphs is reported as ACC-cola and ACC-aesw separately, measuring the acceptability of paragraphs. In addition, to capture the word-gram level transfer accuracy, we adopted GLEU (Napoles et al., 2015) and SARI (Xu et al., 2016) which are commonly used in GEC and text revision tasks.

• **Fluency (FL)** To measure the fluency of the generated paragraphs, we use perplexity (PPL), following the fluency evaluation in Dai et al. (2019) and Cao et al. (2020). We fine-tune a pre-trained GPT-2-Large language model (Radford et al., 2019) on the formal-academic training set and use it to calculate the perplexity in generated examples.

• **Semantic Similarity (SIM)** Following previous benchmark (Krishna et al., 2020), we replace n-gram metrics like BLEU (Papineni et al., 2002) with the subword embedding-based SIM model

(Wieting et al., 2019) to capture the semantic similarity. The similarities between a transferred paragraph and its input are reported as SIM-input, while the similarities between the transferred paragraph and its human rewrite reference are denoted as SIM-gold.

• **BARTScore (BARTS)** BARTScore (Yuan et al., 2021) is a metric that formulates the evaluation of generated text as a text generation task from the pretrained language model BART (Lewis et al., 2019). BARTS can outperform other existing metrics in the evaluation of text from different perspectives, such as fluency, accuracy and integrity. A higher BARTScore indicates that the reference paragraph is more likely generated from the input paragraph using a pre-trained BART model.

### 4.2 Quality of Formal-academic Rewrite

The quality of the parallel dataset consisting of informal-academic paragraphs and corresponding formal-academic rewrites is critically important for evaluation, so we examine this subset from four aspects: 1) academic formality improvement, 2) fluency improvement, 3) semantic similarity and edit distance, and 4) BARTScore. As illustrated in Table 2, the ACC-cola scores on the development and test sets have shown improvements of 2.8 and 2.65, respectively. In the case of ACC-aesw scores, similar upward trends are observed, with boosts to 6.23 and 8.19, respectively. Meanwhile, the PPL of the formal-academic rewrites decreases by 1.88 and 2.79 when compared with the informal-academic paragraphs. The increased academic formality and reduced PPL show that the formal-academic rewrites indeed improve academic formality and fluency. Lastly, DOOLITTLE has a high semantic similarity and low edit distance, implying that the original paragraphs are of good quality and minimal modifications are performed. This shows that academic writing formalization is a challenging task that requires more fine-grained lexical and structural modifications.

### 4.3 Common Mistakes and Types

To understand what types of mistakes are common in informal-academic paragraphs and what edits have been made in the formal-academic rewrite process, we analyze all of the native rewrites in the test set (Figure 1 gives examples with their corresponding native human rewrites). The major error types parallel our objectives of academic formality, as introduced in § 1. And we observe a high per-

| | |
|---|---|
| **Grammar [S]**: When *apply* Naturalistic Driving Film into the design process [...] | |
| **Grammar [T]**: When *applying* Naturalistic Driving Film into the design process [...] | |
| **Spelling [S]**: In this article we have tried to *sumarize* advantages that zebrafish can offer for immunological research. | |
| **Spelling [T]**: In this article we have tried to *summarize* advantages that zebrafish can offer for immunological research. | |
| **Word Choice [S]**: In *a first* analysis, speed behaviour was found to be non-linear according to Figure 2 | |
| **Word Choice [T]**: In *an initial* analysis, speed behaviour was found to be non-linear according to Figure 2 | |
| **Redundancy [S]**: The final flight was *again* a heading control test to verify *once more* the performance of that channel. | |
| **Redundancy [T]**: The final flight was a heading control test to verify once more the performance of that channel. | |
| **Sentence Structure [S]**: We want to show that some solutions of this equation do not exist *in the future*. | |
| **Sentence Structure [T]**: *In the future*, we want to show that some solutions of this equation do not exist. | |

Figure 1: Examples of common types of mistakes in the negative ground truth data. 'S' denotes a source paragraph which is written by a non-native speaker and 'T' denotes the target paragraph written by a native speaker. We highlight the refined part in *blue* and its original expression in *red*.

centage of grammar and spelling errors (46.48%), word choice issues (39.31%), as well as sentence structure changes (14.00%).

• **Grammar and spelling.** This is the most common type of error and is also the easiest to identify. Many sentences contain grammatical errors in subject-verb agreement, verb tense, and capitalization. Additionally, some sentences also contain misspellings or typographical errors.

• **Word choice.** Many sentences are awkwardly phrased. Although the usage of a word may make sense in the original sentence, a more appropriate word may be substituted to make the sentence more fluent. Redundancy is also a problem, as some sentences are overly discursive when describing ideas that can be concisely explained.

• **Sentence structure.** In certain cases, a modifier may be misplaced or in an incorrect order. There are also a number of instances where the sentences are too short or too long. In these cases, it would be better to combine the short sentences and split the long sentences.

## 5 Method

It could be speculated from our dataset analysis results (§ 4) that our DOOLITTLE task is advanced in terms of both quality and difficulty. To address our task with reduced cost and better performance, we proposed a method called metric-oriented reinforcement learning (MORL). This methodology, inspired by reinforcement learning from human feedback (RLHF) (Ziegler et al., 2019; Ouyang et al., 2022), follows a similar three-step training process to RLHF but with crucial modifications: **Step 1:** Train a policy model (usually a PLM) that can meet the requirements of a task. **Step 2:** Select some metrics that can accurately evaluate the quality of how the task has been performed. Build a

reward model that can score a given policy model's output with a scalar. **Step 3:** Optimize the policy against the reward model using reinforcement learning with the proximal policy optimization (PPO) algorithm (Schulman et al., 2017).

The key distinction between RLHF and MORL lies in **Step 2** where RLHF trained a reward model with collected comparison data while MORL utilizes any existing, machine learning-based or not, tuned or plug-and-play evaluation metrics to generate a reward model. Through incorporating a variety of evaluation metrics into the reward model, the cost of implementing MORL becomes flexible and the potential misalignment between human preference and a single metric can be alleviated.

### 5.1 Policy Models

• **Galactica-1.3B** (Taylor et al., 2022) is a decoder-only policy model. We train a Galactica-1.3B model on the paragraph pairs of the format `[[paragraph A]]=[[paragraph B]]` twice. For the first time, paragraph A is sampled from the formal-academic training set of DOOLITTLE, and paragraph B is exactly a duplication of paragraph A. For the second time, paragraph A and paragraph B are a paragraph pair from the development set of DOOLITTLE, where paragraph A is informal-academic and paragraph B is its refined version. After these two stages, the Galactica-1.3B model learns to improve the left paragraph and put the refined result on the right while preserving most of the original content. In the end, after a final prompt of `[[paragraph A]]=`, we sample from the model with beam search (number of beams=4) and extract the content within the second double-bracket-brace as the refined version of paragraph A.

• **BART-Large** (Lewis et al., 2019) is a born strong baseline for GEC task (Katsumata and Komachi, 2020). We selected the BART-Large model with

| | Academic Formality | | | | | Fluency | | Similarity | | | BARTS |
|---|---|---|---|---|---|---|---|---|---|---|---|
| Metric | ACC-cola | ACC-aesw | SARI | GLEU | GPT-4 | PPL | GPT-4 | SIM-input | SIM-gold | GPT-4 | BARTS |
| Input | 95.66 | 69.64 | - | - | 4.32 | 35.97 | 4.55 | - | 98.09 | - | - |
| Style Transfer Models | | | | | | | | | | | |
| ControlledGen | 92.77 | 48.19 | 48.59 | 54.54 | 3.87 | 60.87 | 4.13 | 95.21 | 93.62 | 4.20 | -1.64 |
| DeepLatentSequence | 84.81 | 50.36 | 37.46 | 50.40 | 3.55 | 68.45 | 4.15 | 90.45 | 88.97 | 3.78 | -2.06 |
| StyleTransformer | 85.30 | 56.63 | 38.46 | 50.87 | 3.96 | 66.87 | 4.38 | 90.27 | 88.79 | 3.64 | -2.19 |
| DeleteAndRetrieve | 66.50 | 66.02 | 7.98 | 1.07 | 2.91 | 34.11 | 3.36 | 21.12 | 20.27 | 2.22 | -5.90 |
| GEC Models | | | | | | | | | | | |
| SequentialTransfer | 94.70 | 70.36 | 49.17 | 71.30 | 4.32 | 41.19 | 4.45 | 96.80 | 95.55 | 4.26 | -2.30 |
| BART-GEC | 95.90 | 70.12 | 69.10 | 74.72 | 4.40 | 35.83 | 4.66 | 99.01 | 97.24 | 4.94 | -2.14 |
| Instruction Tuned Models | | | | | | | | | | | |
| ChatGPT | **99.20** | **82.56** | 48.84 | 70.21 | 4.58 | **28.84** | 4.81 | 94.58 | 94.87 | **4.73** | -1.62 |
| MORL-BARTLarge | 97.83 | 78.80 | 55.74 | 75.75 | 4.57 | 35.65 | 4.78 | 98.49 | 97.45 | 4.35 | **-1.32** |
| MORL-Galactica1.3B | 97.83 | 80.24 | **63.79** | **78.37** | **4.60** | 34.50 | **4.86** | **98.72** | **98.30** | 4.70 | -1.34 |
| Native Rewrite | 98.31 | 77.83 | - | - | 4.59 | 33.18 | 4.89 | 98.09 | - | 4.95 | -1.24 |

Table 3: Results of models on DOOLITTLE test paragraphs. Automatic evaluation and GPT-4 judgments of academic formality, fluency, and meaning preservation are reported. The highest scores of each metric among three instruction-tuned models are **bolded**. Some metrics are not applicable for Input and Native Rewrite as they are derived from comparison against these two sets, which are marked by '-'.

406M parameters, fine-tuned it on paragraph pairs in the development set of academic writing formalization, and used the tuned model as the second policy model.

## 5.2 Reward Model

To make a more comprehensive reward model, we amalgamate all four metrics mentioned in § 4.1. For Transfer Accuracy (ACC), instead of using the binary classification result, we utilize the classification logits as the ACC score, and only ACC-aesw is used. For other metrics (PPL, SIM-input, and BARTScore), we directly take the unchanged evaluation results. Each time our policy model generates an output (usually a single paragraph), the reward model first evaluates it utilizing all four metrics. Following this, a weighted sum of all evaluation scores is calculated as the final reward. The weights assigned to each metric are manually determined and optimized through a series of experiments.

## 6 Experiment

### 6.1 Experimental Settings

We apply nine models – four from non-parallel text style transfer (**ControlledGen** (Hu et al., 2017), **DeepLatentSequence** (He et al., 2020), **Style-Transformer** (Dai et al., 2019), and **DeleteAndRetrieve** (Li et al., 2018)), two from low-resource GEC tasks (**SequentialTransfer** (Choe et al., 2019) and **BART-GEC** (Katsumata and Komachi, 2020)), one from ChatGPT, and two MORL-based models

– on DOOLITTLE to establish baseline performance. We use the official code released by the authors of our baseline (except ChatGPT), and follow the recommended configuration settings from their corresponding papers. We report implementation details and hyper-parameter settings of the different models with their size and running speed in the Appendix.

### 6.2 GPT-4 Annotation

GPT-4-based annotation has been proven to be effective in multiple text annotation tasks (Gilardi et al., 2023; Zheng et al., 2023). Considering the advantages of GPT-4 in performance and cost-effectiveness compared to human annotation on MTurk, we apply GPT-4 to evaluate the refinement results of all models on the DOOLITTLE test set. Corresponding to the automatic evaluation metrics, three annotation tasks are assigned to GPT-4 where each focuses on one of the three aspects: Academic Formality, Fluency, and Similarity versus input. For each annotation task, we first feed GPT-4 a comprehensive task description including the task introduction, grading rubrics, the expected input format, and the asked output format. Then, the texts are given as formatted batches of paragraphs, and the evaluation scores are fetched via scripts from GPT-4's response. For each model, we sampled the first 100 paragraphs from its generation results on the informal-academic DOOLITTLE test set. Additionally, we also sampled the first 100 paragraphs

from both formal-academic and informal-academic academic writing formalization test sets, aiding nuanced analyses of each model's performance. These GPT-4 annotation scores are reported with a 5-scale value in Table 3. Appendix E gives the detailed task description.

## 6.3 Overall Performance

Table 3 reports the evaluation results. First, we observe that all models targeted for generic style transfer task — **ControlledGen**, **DeepLatentSequence**, **StyleTransformer** and **DeleteAndRetrieve** — perform much worse than the inputs across all metrics. Second, the results demonstrate that GEC-based models — namely **SequentialTransfer** and **BART-GEC** — outperform style-transfer-based models and yield results that are slightly inferior or comparable to the inputs. This is consistent with our expectation, as simply improving grammar by editing will make only incremental changes to a given sentence. Therefore, style-transfer-based models lack the ability to preserve the paragraphs' original meaning and may make redundant changes which result in poor "refinement". Third, the evaluation outcomes of **DeleteAndRetrieve** and **SequentialTransfer** reveal the misalignment between automatic evaluation metrics and GPT-4 annotation scores, which can be attributed to the inherent limitations of these automated metrics. No single metric is able to perfectly capture the essence of the academic writing formalization task, as each metric is limited in what it measures and therefore lacks the flexibility that humans or GPT-4 possess to holistically evaluate the formality of a given text. Fourth, all reinforcement-learning-based models — **ChatGPT**, **MORL-BARTLarge** and **MORL-Galactica1.3B** demonstrate superior performance in our academic writing formalization (AWF) task, outperforming all other models on almost all metrics. Specifically, when comparing to the DOOLIT-TLE formal-academic test paragraphs, both **ChatGPT** and **MORL-Galactica1.3B** generate competitively good refined paragraphs, comparable with ground truth formal-academic rewrites in terms of academic formality and fluency, but achieve lower scores for similarity versus input and BARTScore. **MORL-BARTLarge** performs slightly inferior to the other two reinforcement-learning-based models, but still largely outperforms all other non-reinforcement-learning-based models as well as the inputs. Considering the substantial size differ-

ence between ChatGPT and MORL-Galactica1.3B (1.3B) or MORL-BARTLarge (406M), our MORL method exhibits remarkable advantages in both performance and cost-efficiency.

In summary, only the three reinforcement-learning-based models demonstrate a clear competency in enhancing the original inputs in terms of both academic formality and fluency metrics. Nevertheless, none of the models consistently surpass the ground truth formal-academic rewrites across all four metrics. This supports the idea that academic writing formalization (AWF) is indeed a difficult task that goes beyond simply correcting grammatical mistakes.

## 6.4 Case Study

To further analyze the generation quality, we examined all input paragraphs together with every baseline's corresponding output. Table 8 shows some representative sample. We observe that while all models can generate complete and fluent text, they also possess specific limitations: **DeleteAndRetrieve** generates the text in a formal-academic way with appropriate sentence structure but struggles with preserving meaning; **ControlledGen**, **DeepLatentSequence**, and **StyleTransformer** can not provide any actual improvements to the input, they either prefer to stay the same as the original input or may modify the original semantics; **SequentialTransfer** and **BART-GEC** can sometimes successfully make necessary changes to make the input more formal-academic — however, most of the time, they are incapable of modifying more grammatical errors; **ChatGPT**, **MORL-BARTLarge**, **MORL-Galactica1.3B** provides a convincing refined paragraph with necessary refinements. However, either some errors are ignored or the sentence's original meaning is changed which results in non-perfect rewrite. It can be clearly observed that all the above models still perform worse than formal-academic human rewrites, thus indicating the difficulty of our academic writing formalization task. Additional cases are included in the Appendix F.

## 6.5 Ablation Study

In this section, we perform several ablations on **MORL-BARTLarge** to study how each metric used in MORL as well as the whole MORL module affects the performance of **MORL-BARTLarge**. In these ablation experiments, we manually set the weight of one metric to zero, and then perform

| Method | ACC-aesw | PPL | SIM-input | SIM-gold | BARTS |
|---|---|---|---|---|---|
| BARTLarge w/o MORL | 74.70 | 38.39 | **99.19** | 96.74 | -1.34 |
| MORL-BARTLarge | **78.80** | 35.65 | 98.49 | 97.45 | -1.32 |
| MORL-BARTLarge w/o ACC | 75.18 | 36.10 | 98.49 | 97.25 | **-1.29** |
| MORL-BARTLarge w/o BARTS | 78.55 | 37.90 | 97.97 | 96.86 | -1.46 |
| MORL-BARTLarge w/o PPL | 77.83 | 41.15 | 98.68 | **97.53** | -1.32 |
| MORL-BARTLarge w/o SIM | **78.80** | **35.61** | 97.74 | 96.67 | -1.44 |

Table 4: Ablation studies of MORL-BARTLarge models. BARTLarge w/o MORL denotes the BART-Large policy model without MORL tuning. MORL-BARTLarge w/o denotes that the corresponding metric's weight is set to zero during MORL-tuning.

MORL tuning on the BART-Large policy model described in § 5.2 with all other parameters the same as our optimized best setting. Thus, we can study the contribution of each individual metric to the **MORL-BARTLarge** model. These results are shown in Table 4.

Comparing the results of the BART-Large policy model with and without MORL-tuning, we can tell a noticeable performance gain across most of the automatic metrics except SIM-input, which indicates that MORL effectively boosts the performance of the given policy model. Those variants of **MORL-BARTLarge** models, namely MORL-BARTLarge w/o, produced the worst evaluation score of a specific metric among all MORL-tuned models without exception. This phenomenon reveals that removing a certain metric from the MORL-tuning process will prevent MORL from optimizing the policy model's properties which can be measured by that metric. Each metric incorporated into MORL is practical in helping MORL improve the policy model against the metric itself.

## 7   Conclusion and Future Work

We propose a new setting for language refinement called Academic Writing Formalization (AWF), which bridges the gap between formal-academic and informal-academic writing. We contribute a new dataset, DOOLITTLE, and evaluate nine baseline models for AWF using both automatic metrics and GPT-4 annotation, demonstrating that paragraph-level refinement is a promising task with significant room for improvement.

To address AWF, we propose a method called metric-oriented reinforcement learning (MORL). Leveraging MORL, we successfully elevate the performance of BART-Large and Galactica-1.3B, yielding results comparable to those of GPT-3.5 turbo-based ChatGPT, which possesses significantly more parameters than our MORL-based

models. Even though such models do not outperform the formal-academic paragraphs due to the difficulty of academic writing formalization, this promising result has substantiated the advantages in terms of competence and cost-efficiency of MORL.

In the future, we plan to incorporate external knowledge and weakly supervised information into the text rewriting process. We also assert that the full potential of MORL as applied to large language models remains to be tapped. It is foreseeable that through the inclusion of more relevant metrics and advanced modeling, MORL can enhance its capability further and be adapted to a broader spectrum of NLP tasks.

## 8   Limitations

First, as a dataset constructed based on Semantic Scholar Open Research Corpus (S2ORC) (Lo et al., 2020), DOOLITTLE inevitably inherits most limitations that have been discovered or potentially exist from its parent. This is especially true for the non-parallel training set where we do not collect any new data. For those who want to adopt our dataset in the future, a comprehensive study of S2ORC is needed. Second, for the evaluation of model performance, we observed the differences in some results between well-performing models — ChatGPT, MORL-BARTLarge, and MORL-Galactica1.3B — are indeed subtle. Considering the randomness in language model generation, our current evaluation metrics lack discriminability on our AWF task. Hence, we only draw indicative conclusions about the models' performance, such as "They are competitively good...", lacking more concrete details. Third, prompt tuning has been applied several times in this paper. However, it is not possible to determine the most suitable prompt from the infinite space of prompt choices. Thus, despite all the prompts we mentioned being properly tuned from a series of experiments, we do not guarantee that our

methods fully exploit the potential of each model and the capability of GPT-4 annotation.

## 9 Ethical Considerations

The data and task presented in this paper are of a sensitive nature, as there is the concern of judging the English writing of scientific researchers. We acknowledge this and highlight some of the measures we have taken below.

• Anonymity. We are aware that it is possible to use a search engine to de-anonymize paragraphs. Our preliminary investigations yield that over 40% of our data do not yield any results on Google. We also deliberately exclude metadata of the paragraphs to further protect the authors' identity. While acknowledging the potential of the dual use of DOOLITTLE, we believe that our dataset, when not actively abused and extensively searched on Google, can be useful for style transfer tasks. It would be a misuse of DOOLITTLEto identify authors and use their traits to correlate with mastery of academic writing.

• Potential Error. Given the nature of our dataset construction with human annotators, there will be errors and disagreements in the annotation: certain academic formality scores are inaccurate. We emphasize that these scores are not to be taken as any form of absolute judgment on any scholars' English writing ability.

• Academic Formality Classification. We present this dataset not only with the purpose of exploring academic formality classification, but also as an extension to the task of writing style transfer. As such, this task can be broadened to other types of text style transfer. To further focus on the academic formality task, it is also possible for one to construct a dataset of papers to emulate and use as positive examples.

• Copyright. The S2ORC dataset is constructed from open-access academic papers without any copyright issues. Note that there is no licensing information that accompanies the original S2ORC dataset[3]. We thus infer that third-parties are allowed to use the data in view of the open-access status of the papers.

In light of these issues, we plan to follow a data release plan where DOOLITTLE will only be released to researchers who explicitly consent to not de-anonymize any of the paragraphs.

---

[3]https://github.com/allenai/s2orc/

## Acknowledgments

We thank the anonymous reviewers for their valuable suggestions. This work was supported by the General Research Fund (GRF) of Hong Kong (No. 16310222). Shizhe Diao was supported by the Hong Kong Ph.D. Fellowship Scheme (HKPFS).

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

# Appendix

## A Annotation Details

### A.1 How to Determine Academic Formality

One potential avenue is to distinguish academic formality by the country of the first author's affiliation. However, we eschew this method for three reasons: 1) The first author does not necessarily contribute to all of the paper writing. 2) Even in primarily English-speaking countries, there are many who do not speak English as their first language. 3) Determining academic formality by country involves the ethical issue of regional discrimination. Instead, we adopt an unbiased means that ignores the paper authors and instead asks human annotators to rate the academic formality of each paragraph via crowdsourcing.

### A.2 Annotator's Instruction

The detailed descriptions are shown below to illustrate the meaning of different scores, which are presented with examples to annotators as well.

- **Score 0 [incomplete]**. For any incomplete sentences, which is only a sentence fragment, you may give it 0 score.
- **Score 1 [informal-academic]**. You are 100% sure that the paragraph was written by a non-native writer. The language expression is very unclear or weird that you have no idea what meaning he is trying to express even by guessing. Usually, there are serious grammatical errors, spelling problems. Redundancy problems.
- **Score 2 [somewhat informal-academic]**. You are not 100% sure that the paragraph was written by a non-native writer. The language expression is unclear that you cannot fully understand the meaning. However, you can guess what meaning he/she is trying to express. Usually, the paragraph is fine with slight errors. For example, there are some spelling, punctuation, capitalization problems. Redundancy problem, which means there could be a better expression, for example, split a long sentence into two short sentences.
- **Score 3 [between formal-academic and informal-academic]**. You are not confident whether the paragraph was written by a non-native writer or not. You can fully understand the meaning he/she is trying to express without guessing. However, the language expression is rigid and unnatural (for example, Chinglish). A native writer usually won't express the same meaning in this way.
- **Score 4 [somewhat formal-academic]**. You are not 100% sure that the paragraph was written by a native writer. You can fully understand the meaning the paragraph is trying to express and the language is natural and fluent. The language expression is very consistent with the style of native writers. However, there are minor parts in the paragraph that can be further improved to make it closer to formal-academic English. For example, replace a certain word with another word to express the meaning more precisely.
- **Score 5 [formal-academic]**. You are 100% sure that the paragraph was written by a native writer, which means you cannot rewrite it better than it. The language is very clear and fluent, completely in the style of native writers. You cannot rewrite a better one.

### A.3 Gold Set Construction

Prior to large-scale annotation, four experts — two of whom are native English speakers and the other two are authors of this paper (one is a native English speaker) — worked together to produce a gold set for quality control. The task is the same as the one introduced before, and the annotation guidelines are presented to the annotators. We provide sufficient training for these two external annotators, inclusive of discussion, to ensure consistency and quality. Conflicting annotations were discussed by all four annotators to produce a final rating. In the end, we construct a gold set consisting of 500 paragraphs as the probing set, which is injected into the large-scale annotation tasks for quality assurance in the crowdsourced annotation.

### A.4 Analysis of Annotation

Our data source, S2ORC, contains a diverse set of academic disciplines, whose resultant discipline distribution is shown in Figures 2 and 3. We draw two observations, both quite consistent with the overall distribution of the

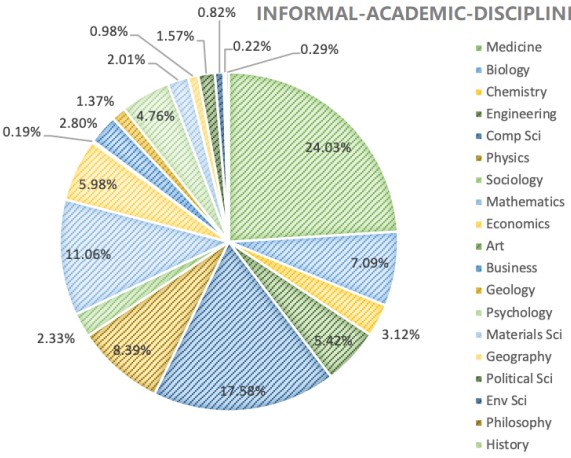

Figure 2: Disciplines on informal-academic dataset

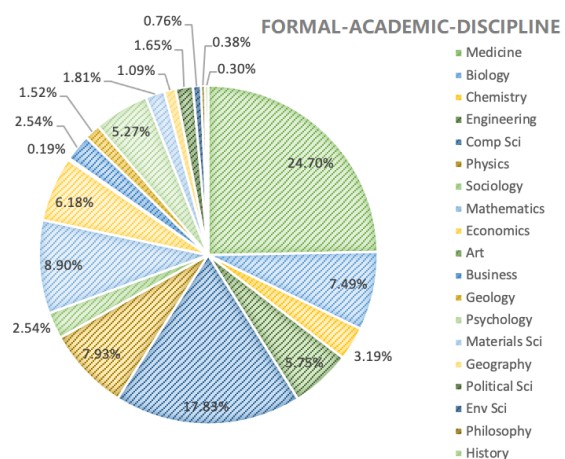

Figure 3: Disciplines on formal-academic dataset

original dataset, S2ORC: 1) There are 19 disciplines in total and the distribution is similar between the formal-academic and informal-academic datasets. 2) Medicine, Computer Science, Mathematics, Physics, and Biology are the five top fields of study.

We also analyze the difference between preprints without peer review and published papers after peer review. Using the metadata provided by S2ORC, we can infer that a paper was published if it possesses an ACL ID (the unique ID for papers on the ACL Anthology), a DOI (Digital Object Identifier), or venue/journal information. Our results indicate that 83.5% of published paragraphs are identified as formal-academic by annotators while this ratio is 81.1% for preprints. We also evaluate the academic formality score and PPL for these two splits and observe that published paragraphs have a higher academic formality score (82.80 versus 78.31) and lower PPL (24.22 versus 26.47), demonstrating that published paragraphs sound more native and fluent. This is reasonable as authors make extensive revisions after the review process and address many writing issues following the feedback from reviewers. Therefore, peer review is a key process for improving writing quality.

## B Experimental Settings

We apply nine models – four from non-parallel text style transfer, two from low-resource GEC tasks, ChatGPT, and two MORL-based models – on DOOLITTLE to establish a baseline performance. Other than our proposed method, these

approaches are chosen as 1) they are highly related to our proposed academic formality transfer task; 2) they require non-parallel data or only limited parallel data. 3) ChatGPT is the current SOTA Chatbot whose performance in generic NLP tasks has been proven. Hence, they provide a good starting point for approaching our proposed task. By analyzing our experimental results, we highlight the challenges inherent in this task and provide insights into future research directions. In this section, we provide the details of baseline models and training settings.

### B.1 Baseline Models

• **ControlledGen** (Hu et al., 2017): a model combining variational auto-encoders (VAEs) and attribute discriminators to learn disentangled latent representations with designated semantics.

• **DeepLatentSequence** (He et al., 2020): a generative probabilistic model with few independence assumptions based on a standard attentional sequence-to-sequence approach and an encoder-decoder architecture.

• **StyleTransformer** (Dai et al., 2019): a Transformer-based model for learning content and style vectors without parallel data by cyclic reconstruction.

• **DeleteAndRetrieve** (Li et al., 2018): an RNN-based model which firstly extracts content words by removing style-dependent phrases and then retrieves and integrates new phrases related to the target attribute into a fluent sentence.

- **SequentialTransfer** (Choe et al., 2019): a low-resource GEC method using a realistic noising function to generate synthetic parallel corpora which are applied to pre-train a Transformer model. Then the pre-trained model is adapted to the targeted dataset by fine-tuning.
- **BART-GEC** (Katsumata and Komachi, 2020): a baseline model from GEC which utilizes BART (Lewis et al., 2020) as a pretrained model and fine-tunes the model on the target dataset.
- **ChatGPT** (OpenAI, 2022): a chatbot developed by OpenAI based on their large language model GPT-3.5-Turbo (Brown et al., 2020) which can handle a variety of text generation tasks in a question-Answering fashion with properly adjusted prompts.

To adapt **ControlledGen**, **DeepLatentSequence**, **StyleTransformer**, and **DeleteAndRetrieve** to our task, we simply treat formal-academic and informal-academic as two different styles and train the models by following the text style transfer pipeline with our non-parallel training data. For **SequentialTransfer**, we follow Choe et al. (2019) to use a noising function on several high-quality corpora as well as our formal-academic training data to generate synthetic parallel data in order to pre-train the transformer-based model. Then we use parallel data from the development set to fine-tune the model. For **BART-GEC**, we follow Katsumata and Komachi (2020) to use BART as a pre-trained model and fine-tune the model using parallel data from the development set. For **ChatGPT**, we tested a variety of question templates and manually select the one that can make **ChatGPT** perform the best in our AWF task, which is mentioned in Appendix D. For **MORL-BARTLarge** and **MORL-Galactica1.3B**, we first fine-tune two policy models following instructions in § 5.1 from the pretrained models. Then, we optimize those tuned policy models using our reward model mentioned in § 5.2 with the PPO algorithm implemented through Transformer Reinforcement Learning (TRL) library (von Werra et al., 2020). For **MORL-BARTLarge**, we input raw paragraphs of DOOLITTLE formal-academic development set to the policy model and feed the raw sampled output to the reward model to get its scalar reward. For **MORL-Galactica1.3B**, we also use DOOLITTLE formal-academic development set as the input data source. However, instead of directly feeding raw paragraphs to the policy model, we first pre-process the input paragraph to the format described in § 5.1— `[[ raw paragraph ]] =`. In the end, we extracted the paragraph within the second double-bracket-brace from the raw generated output as the input to the reward model. One more thing to mention is that, during the training process of MORL, we also calculate the KL-divergence between outputs from policy models before and after reinforcement-learning-optimization to ensure the optimized model does not deviate too much from the original one.

### B.2 Hyper-parameter Settings

Table 5 reports the hyper-parameters we used for tuning our baselines and our models tuned with MORL. For each model, we try combinations of the hyper-parameters and report the one with the highest academic formality score in our paper. Each model is trained on a Tesla V100S-PCIE GPU with 32GB memory. Table 6 reports the hyperparameter configurations for best-performing models of the baseline models.

| Types | Values |
|---|---|
| Learning Rate | $10^{-6}, 10^{-5}, 3 \times 10^{-5}, 10^{-4}$ |
| Dropout Rate | $0.1, 0.2, 0.3, 0.4, 0.5$ |
| Batch Size | $1, 4, 8, 16, 32$ |
| Embedding Dimensions | $128, 256, 300, 512, 768$ |
| Max Input Length | $100, 130$ |
| Metric Weight | $2 \times 10^{-4}, 5 \times 10^{-3}, 0.1, 1$ |

Table 5: The hyper-parameters for tuning our baselines where Metric Weight is only applicable for MORL tuning.

## C   Model Size and Running Speed

Table 7 reports the number of trainable parameters and the inference speed (sentences/second) of all models except ChatGPT on the benchmark. The test is performed on Tesla V100S-PCIE GPU with 32GB memory.

## D   Description of ChatGPT AWF Task

|  | CG | DLS | ST | DAR | SQ | BA | MB | MG |
|---|---|---|---|---|---|---|---|---|
| Max epochs | 30 | 5 | 5000 | 500 | 30 | 1000 | 5 | 5 |
| Best epochs | 10 | 4 | 2500 | 300 | 15 | 500 | 2 | 5 |
| Learning Rate | $3 \times 10^{-4}$ | $10^{-3}$ | $10^{-4}$ | $3 \times 10^{-4}$ | $3 \times 10^{-4}$ | $3 \times 10^{-5}$ | $10^{-6}$ | $10^{-6}$ |
| Dropout Rate | 0.5 | 0.3 | 0.1 | 0.3 | 0.3 | 0.3 | 0.1 | 0.1 |
| Batch Size | 16 | 32 | 16 | 16 | 32 | 32 | 4 | 1 |
| Embedding Dimensions | 300 | 128 | 256 | 128 | 512 | 1024 | 1024 | 2048 |
| Max Input Length | 100 | 130 | 100 | 100 | 100 | 100 | 128 | 256 |

Table 6: The hyperparameter configurations for best-performing models of all models except ChatGPT. CG, ST, DAR, SQ, BA, MB and MG denote ControlledGen, StyleTransformer, DeleteAndRetrieve, SequentialTransfer, BART-GEC, MORL-BARTLarge and MORL-Galactica1.3B.

| Dataset | CG | | DLS | | ST | | DAR | | SQ | | BA | | MB | | MG | |
|---|---|---|---|---|---|---|---|---|---|---|---|---|---|---|---|---|
| | P. | S. | P. | S. | P. | S. | P. | S. | P. | S. | P. | S. | P. | S. | P. | S. |
| Test dataset | 141M | 4.76 | 69M | 2.23 | 134M | 0.97 | 100M | 0.67 | 123M | 5.49 | 406M | 2.13 | 406M | 2.13 | 1.3B | 0.62 |

Table 7: The number of trainable parameters (P.) and the running speed (sentences/second, S.) on the test sets of all models except ChatGPT. CG, ST, DAR, SQ, BA, MB, and MG denote ControlledGen, StyleTransformer, DeleteAndRetrieve, SequentialTransfer, BART-GEC, MORL-BARTLarge, and MORL-Galactica1.3B.

## D.1 ChatGPT AWF Task

Help me refine the following paragraph within "« »" to make it more formal-academic. Specifically, you should follow the 3 steps below:
Step1: Find and locate all grammatic mistakes in terms of grammar and spelling, word choice and sentence structure in the given paragraph. (Note that it is OK if you can't find any mistake. In this case, just respond with the original given paragraph within "« »".)
Step2: Correct all mistakes you found in Step 1 without changing any parts else in the paragraph.
Step3: Output the refined paragraph within "« »" in your response without anything else.

Notice:
1) minimal changes to the original paragraph are preferred
2) you should try to preserve the given paragraph's original meaning and sentence structure as much as possible.

Here are some examples:

example 1
example 2
example 3

The given paragraph is: « paragraph content »

# E Description of GPT-4 Annotation Task

## E.1 Academic Formality Annotation Task

You are asked to participate in a text evaluation task whose main objective is to score the degree of "Academic Formality" for given paragraphs. Evaluating the degree of "Academic Formality" means judging whether a given paragraph sounds like a paragraph written by a native English speaker or not. The scale of Academic Formality score is an integer from 1 to 5. The detailed scoring rubric is as below:

Score 0: If the paragraph contains any incomplete sentence which doesn't sounds likes being written by human.
Score 1: You are 100% sure that the paragraph was written by a non-native speaker. The language expression is very unclear or weird which makes you have no idea what meaning the author is trying to express even by guessing.
Score 2: You think the paragraph was probably written by a non-native speaker. The language expression is unclear that you cannot fully understand the meaning. But you can guess what meaning the paragraph is trying to express.
Score 3: You are not confident about whether the paragraph was written by a non-native speaker or not. You can fully understand what the paragraph is trying to express without guessing. However, the language expression is rigid and unnatural (for example Chinglish). A native English speaker usually won't express the same meaning in this way.
Score 4: You think the paragraph was probably written by a native speaker. You can fully understand the meaning the paragraph is trying to express and the language is natural and fluent. The language expression is very consistent with the style of native English speakers. But there are minor parts in the paragraph that can be further improved to make it closer to native English. For example, replace a certain word with another word to express the meaning more precisely.
Score 5: You are 100% sure that the paragraph was written by a native speaker, which means you cannot rewrite it any better. The language is very clear and fluent, completely in the style of native English speakers. You cannot rewrite a better one.

Each time, you will be given a batch of 20 paragraphs with format below:
Paragraph 1: <Content of Paragraph 1>
Paragraph 2: <Content of Paragraph 2>
······
Paragraph 20: <Content of Paragraph 20>
You should only output a json object that contains the following keys: Paragraph ID, Scoring Reason and Academic Formality Score. Note for "Scoring Reason", you need to briefly elaborate the reason why you grade the given paragraph such a Academic Formality score within 20 words.

## E.2 Fluency Annotation Task

You are asked to participate in a text evaluation task whose main objective is to score the degree of "Fluency" for given paragraphs. Evaluating the degree of "Fluency" means judging whether the paragraph is consistent and coherent. The scale of Fluency score is an integer from 1 to 5. The detailed scoring rubric is as below:

Score 0: If the paragraph contains any incomplete sentence which doesn't sounds likes being written by human.
Score 1: The paragraph is neither consistent nor coherent at all which makes you have no idea about what the paragraph is trying to express even by guessing. Usually, it means some parts of the paragraph are not related to others. Note that a paragraph with errors like logic bugs and contradictions should not be scored to 1 since the occurrences of these errors require the context has some relation.
Score 2: The paragraph is neither consistent nor coherent, but you can still have a brief idea about what the given paragraph is trying to express by guessing. The only difference between scores 1 and 2 is that for score 2, different parts of the given paragraph are more or less related to others. However, you can only guess at what the paragraph is trying to express as these parts are managed without logic. Score 3: You can fully understand what the given paragraph is trying to express without guessing. However, it is obvious that the content is neither coherent nor consistent. For example, you can easily find a contradiction within the paragraph even though you are not an expert in this discipline. For these paragraphs, you should be able to rewrite them by adding, deleting, or exchanging a few words but hard to keep its original meaning.
Score 4: You can easily understand what the given paragraph is trying to express since its content is both coherent and consistent. However, you can still find some minor contradictions or bugs that can only be found by experts or native English speakers. For these paragraphs, you should be able to rewrite it to make it more consistent or coherent by adding, deleting, or exchanging few words while keeping its original meaning. Score 5: The content of the given paragraph is both coherent and consistent. For these paragraphs, you can't rewrite them to make them more consistent or coherent.

Each time, you will be given a batch of 20 paragraphs with format below:
Paragraph 1: <Content of Paragraph 1>
Paragraph 2: <Content of Paragraph 2>
……
Paragraph 20: <Content of Paragraph 20>

You should only output a json object that contains the following keys: Paragraph ID, Scoring Reason and Fluency Score. Note for "Scoring Reason", you need to briefly elaborate the reason why you grade the given paragraph such a Fluency score within 20 words.

## E.3 Similarity Annotation Task

You are asked to participate in a text evaluation task whose main objective is to score the degree of "Similarity" for given paragraphs and their respective reference paragraphs. Evaluating the degree of "Similarity" means to judge whether the given paragraph is similar to its reference paragraph in terms of content, vocabulary usage and writing style.

The scale of Similarity score is an integer from 1 to 5. The detailed scoring rubric is as below:
Score 0: If the given paragraph contains any incomplete sentence which doesn't sounds likes being written by human. In this case, you can directly give the given paragraph score 0 without reading its reference paragraph.
Score 1: The content of the given paragraph has nothing to do with its reference paragraph. The things they are trying to express don't even belong to the same discipline or area of research. The sentence structure of given paragraph is not similar to its reference at all.
Score 2: You can distinguish the given paragraph and its reference paragraph are talking about two unrelated things. However, you can tell what they are trying to express belong to a same discipline or area of research because they share some similarities in one or two of the 3 aspects: vocabulary usage, writing style or sentence structure.
Score 3: You can distinguish the given paragraph and its reference paragraph are talking about two unrelated things. However, you can tell what they are trying to express belong to a same discipline or area of research because they share some similarities in all 3 aspects: vocabulary usage, writing style and sentence structure.
Score 4: You can determine that the given paragraph and its reference paragraph are talking about a same thing and they share similar vocabulary and sentence structure. However, they may not share the same viewpoint or focus on the same aspect of that thing.
Score 5: The given paragraph and its reference are very similar, and they are talking about the exact same thing with exact same viewpoint and focus. In this case, you can only find few differences between the given paragraph and its reference, like some words being replaced with its synonyms, differences in tenses or some minor grammatic errors etc.

Each time, you will be given a batch of 10 paragraphs and their respective reference paragraphs with format below:

Paragraph 1: <Content of Paragraph 1>
Reference paragraph for paragraph 1: <Content of Paragraph 1's reference paragraph>
Paragraph 2: <Content of Paragraph 2>
Reference paragraph for paragraph 2: <Content of Paragraph 2's reference paragraph>
……
Paragraph 10: <Content of Paragraph 10>
Reference paragraph for paragraph10: <Content of Paragraph 10's reference paragraph>

You should only output a json object that contains the following keys: Paragraph id, Scoring Reason and Similarity Score. Note for "Scoring Reason", you need to briefly elaborate the reason why you grade the given paragraph such a Similarity score within 20 words.

Response "Yes, I'm ready" if you fully understand your task.

## F  Examples for Case Study

Table 8 shows three more generated samples from our baseline models.

| Model | Paragraph Example |
|---|---|
| **Origin** | In each *subsections* the effect of each *parameters* are analyzed *and* individually. |
| **ControlledGen** | In each *subsections* the effect of each *parameters* are analyzed *and* individually. |
| **DeepLatentSequence** | *In each way* the effect of each *parameters* are analyzed *and* individually . |
| **StyleTransformer** | *These results are explained in details in the subsequent subsections.* |
| **DeleteAndRetrieve** | *These results show of the best parameter values for the corresponding [...]* |
| **SequentialTransfer** | In each *subsection* the effect of each *parameter* are analyzed *and* individually. |
| **BART-GEC** | In each *subsections,* the effect of each *parameters* are analyzed *and* individually. |
| **ChatGPT** | In each *subsection,* the *effects* of each *parameter are analyzed individually.* |
| **MORL-BARTLarge** | In each *subsection,* the *effects* of each *parameter are* analyzed *and* individually. |
| **MORL-Galactica1.3B** | In each *subsection,* the effect of each *parameter is analyzed and* individually. |
| **GroundTruth** | In each *subsection,* the effect of each *parameter is analyzed individually.* |
| **Origin** | When *apply* Naturalistic Driving Film *into* the design process, there are several aspects need to *take* into consideration. |
| **ControlledGen** | When *apply* Naturalistic Driving Film *into* the design process, there are several aspects need to *take* into consideration. |
| **DeepLatentSequence** | When apply *longitudinally* driving film into the design process, there are several aspects need to take into consideration. |
| **StyleTransformer** | When apply *Realist* Driving Film into the design process, there are several aspects need to *take* into consideration. |
| **DeleteAndRetrieve** | *When the student's interest is the same as the most important thing ...* |
| **SequentialTransfer** | When *applying* Naturalistic Driving Film into the design process, there are several aspects needed to *take* into consideration. |
| **BART-GEC** | When *apply* Naturalistic Driving Film into the design process, there are several aspects need to *take* into consideration. |
| **ChatGPT** | When *incorporating* Naturalistic Driving Film *into* the design process, there are several aspects that need to *be taken* into consideration. |
| **MORL-BARTLarge** | When *apply* Naturalistic Driving Film *into* the design process, there are several aspects to *need to* take into consideration. |
| **MORL-Galactica1.3B** | When *apply* Naturalistic Driving Film *into* the design process, there are several aspects need to *be considered*. |
| **GroudTruth** | When *applying* the Naturalistic Driving Film *in* the design process, there are several aspects that need to *be taken* into consideration. |
| **Origin** | ... of histamine biosynthesis would *be expected to* be a useful tool in the analysis of the biological roles of this biogenetic amine. |
| **ControlledGen** | ... of histamine biosynthesis would *be expected to* be a useful tool in the analysis of the biological roles of this *nearfields* amine. |
| **DeepLatentSequence** | ... of gabaergic biosynthesis would *be expected to* be a useful tool in the analysis of the biological roles of this *enigmatic* entity. |
| **StyleTransformer** | ... of histamine biosynthesis would *be expected to* be a useful tool in the analysis of the biological roles of this *pathogenesis*. |
| **DeleteAndRetrieve** | *... present the results of the study of the different types of remifentanil.* |
| **SequentialTransfer** | ... of histamine biosynthesis would *be expected to* be a useful tool in the analysis of the biological roles of this *biogenic* amine. |
| **BART-GEC** | ... of histamine biosynthesis would *be expected to* be a useful tool in the analysis of the biological roles of this biogenetic amine. |
| **ChatGPT** | ... of histamine biosynthesis would *be expected to* be a useful tool in the analysis of the biological roles of this *biogenic* amine. |
| **MORL-BARTLarge** | ... of histamine biosynthesis would be *be expected to* a useful tool in the analysis of the biological roles of this biogenetic amine. |
| **MORL-Galactica1.3B** | ... of histamine biosynthesis would be *useful* in the analysis of the biological roles of this biogenetic amine. |
| **GroundTruth** | ... of histamine biosynthesis would be *be expected to* a useful tool in the analysis of the biological roles of this biogenetic amine. |

Table 8: Examples of refined results and academic writing formalization test set. *Red and italic words* indicates wrong usage in the original sentence, *Blue and italic words* are words refined by insertion or replacement while *Green and italic words* with strikethrough are words regined by deletion.