# OpenReview forum: "Doolittle: Benchmarks and Corpora for Academic Writing Formalization"
_EMNLP/2023/Conference — EMNLP 2023 Main_

### Official Review · Reviewer_rcwz · 2023-08-04

**Soundness:** 3

**Excitement:**

3: Ambivalent: It has merits (e.g., it reports state-of-the-art results, the idea is nice), but there are key weaknesses (e.g., it describes incremental work), and it can significantly benefit from another round of revision. However, I won't object to accepting it if my co-reviewers champion it.

**Paper Topic And Main Contributions:**

This paper proposes the new task of Academic Writing Formalization that aims to generalize the scope of GEC for language refinement, which involves grammar correction, word refinement and structure modification. And it proposes a large-scale, non-parallel dataset named DOOLITTLE with around 800 expert-annotated parallel paragraphs as the development and test sets. Based on the proposed dataset, the paper proposes a method, metric-oriented reinforcement learning (MORL) and conducts a comprehensive evaluation of neural approaches on AWF. The results show that their performance still suffers from a sizable gap compared to formal-academic rewrites by humans.

**Reasons To Accept:**

1. The paper proposes the AWF task and constructs the large-scale dataset DOOLITTLE, which is beneficial for improving the overall quality of formal academic writing and achieving automation.
2. The paper provides a detailed description of the annotation process and quality control of the dataset and it is helpful for development of future related work.
3. The paper conducted extensive experiments, and these models can serve as a baseline for future work.


**Reasons To Reject:**

1. The paper did not provide a clear explanation of the motivation for selecting automatic evaluation metrics. Observing a high proportion of grammar and spelling errors in the test set, It seems necessary to introduce GLUE or F0.5 commonly used in GEC, as well as SARI commonly used in text revisions. But these metrics are not presented in the paper.
2. As the proposed dataset is non-parallel, it does not support training the model in the Seq2Seq mode, which to some extent limits the universality of the dataset.
3. The paper proposes the MORL method. However, due to the lack of sufficient ablation experiments, we cannot know how much gain MORL has achieved.
4. The analysis of the experimental results could to be further in-depth. For example, what is the reason for the significant differences in PPL between different models?
5. Some details in the paper still need to be considered. For example, although there is currently no clear definition of LLM, it is generally believed to be a decoder only model above 1B. It seems inaccurate to include BART-Large in the category of LLM.


**Reproducibility:**

4: Could mostly reproduce the results, but there may be some variation because of sample variance or minor variations in their interpretation of the protocol or method.

**Reviewer Confidence:**

3: Pretty sure, but there's a chance I missed something. Although I have a good feel for this area in general, I did not carefully check the paper's details, e.g., the math, experimental design, or novelty.

---

> ### Author Rebuttal · Authors · 2023-08-29
>
> Dear Reviewer rcwz,
>
> Thank you very much for your comprehensive review and valuable feedback! We address your comments one by one as follows:
>
>
> **[Automatic evaluation metrics]**
>
> > The paper did not provide a clear explanation of the motivation for selecting automatic evaluation metrics. Observing a high proportion of grammar and spelling errors in the test set, It seems necessary to introduce GLUE or F0.5 commonly used in GEC, as well as SARI commonly used in text revisions. But these metrics are not presented in the paper.
>
> Thanks for your suggestions.
> Following [1], we applied metrics that can evaluate the transfer accuracy, semantic similarity and fluency properties of baseline style-transfer/GEC models. We did not use the n-gram-based metrics due to their inherent limitations elaborated in [1], including:
>
> (1)unreliable correlations between n-gram overlap and human evaluations of semantic similarity
>
> (2)discouraging output diversity
>
> However, to provide a more comprehensive analysis, following your advice, we include SARI and GLEU for our task with more evaluation results. The evaluation results of SARI and GLEU on all baselines are shown below. We find that our proposed method MORL achieves significant improvements over baseline methods. Thanks for your question and we have included these new results in our revised manuscript.
>
> | | SARI | GLEU |
> | ----------- | ----------- | ----------- |
> | ControlledGen | 48.59 | 54.54 |
> | DeepLatentSequence | 37.46 | 50.40 |
> | StyleTransformer | 38.46 | 50.87 |
> | DeleteAndRetrieve | 7.98 | 1.07 |
> | SequentialTransfer | 49.17 | 71.3 |
> | BART-GEC | 69.10 | 74.72 |
> | ChatGPT | 48.84 | 70.21  |
> | MORL-BARTLarge | 55.74 | 75.75 |
> | MORL-Galactica1.3B | 63.79 | 78.37 |
>
>
> [1] Krishna, K., Wieting, J., & Iyyer, M. (2020). Reformulating unsupervised style transfer as paraphrase generation.
>
>
>
> **[Non-parallel data]**
>
> > As the proposed dataset is non-parallel, it does not support training the model in the Seq2Seq mode, which to some extent limits the universality of the dataset.
>
> Thanks for your comments! Considering the high cost of annotating large-scale parallel data,  we hope to explore enhancing model performance and representation using non-parallel data.
> As mentioned in our paper, there are nearly 70K paragraphs in the Formal-academic and Informal-academic training set of DOOLITTLE, which means we need to spend around 70 times more on annotating the full dataset compared with our current work. Thus, considering the cost issue, we propose DOOLITTLE with a non-parallel training set and parallel development and test sets.
> This shares a similar goal with recent non-parallel style transfer [2-4] and low-resource GEC [5-7] studies which aim to build systems that do not rely on large amounts of parallel data.
>
> We have included the discussions and cited these studies in our revised manuscript. Thanks for your question!
>
> [2] Shen, T., Lei, T., Barzilay, R., & Jaakkola, T. (2017). Style transfer from non-parallel text by cross-alignment.
>
> [3] Hu, Z., Yang, Z., Liang, X., Salakhutdinov, R., & Xing, E. P. (2017). Toward controlled generation of text.
>
> [4] Fu, Z., Tan, X., Peng, N., Zhao, D., & Yan, R. (2018). Style transfer in text: Exploration and evaluation.
>
> [5] Bryant, C., Felice, M., Andersen, Ø. E., & Briscoe, T. (2019). The BEA-2019 shared task on grammatical error correction.
>
> [6] Choe, Y. J., Ham, J., Park, K., & Yoon, Y. (2019). A neural grammatical error correction system built on better pre-training and sequential transfer learning.
>
> [7] Grundkiewicz, R., Junczys-Dowmunt, M., & Heafield, K. (2019). Neural grammatical error correction systems with unsupervised pre-training on synthetic data.
>
>
> **[Ablation studies]**
>
> > The paper proposes the MORL method. However, due to the lack of sufficient ablation experiments, we cannot know how much gain MORL has achieved.
>
> Thanks for your suggestion! We agree that adding ablation experiments would enhance the persuasiveness of our research. Therefore, we conduct ablation studies for our proposed method MORL. The results are shown below. We can see that after removing a certain metric, that metric will not be optimized, so the corresponding score will decrease compared to the best result.
>
> |  | ACCaesw | PPL | SIM-input | SIM-gold | BARTS |
> | ----------- | ----------- | ----------- | ----------- | ----------- | ----------- |
> | BARTLarge w/o MORL | 74.70 | 38.39 | 99.19 | 96.74 | -1.34 |
> | MORL-BARTLarge | 78.80 | 35.65 | 98.49 | 97.45 | -1.32 |
> | MORL-BARTLarge w/o ACC | 75.18 | 36.10 | 98.49 | 97.25 | -1.29 |
> | MORL-BARTLarge w/o BARTS | 78.55 | 37.9 | 97.97 | 96.86 | -1.46 |
> | MORL-BARTLarge w/o PPL | 77.83 | 41.15 | 98.68 | 97.53 | -1.32 |
> | MORL-BARTLarge w/o SIM | 78.80 | 35.61 | 97.74 | 96.67 | -1.44 |
> Table 8. Ablation studies of MORL-BARTLarge models. BARTLarge w/o MORL denotes the BART-Large policy model without MORL tuning. MORL-BARTLarge w/o denotes that the corresponding metric’s weight is set to zero during MORL-tuning.
>
> Below is a section for ablation study which we’ll add to our updated manuscript.
>
>
> **Ablation Study**
>
> In this section, we perform several ablations on MORL-BARTLarge to study how each metric used in MORL as well as the whole MORL module affects the performance of MORL-BARTLarge. In these ablation experiments, we manually set the weight of one metric to zero, and then perform MORL tuning on the BART-Large policy model described in section 5.2 with all other parameters the same as our optimized best setting. Thus, we can study the contribution of each individual metric to the MORL-BARTLarge model. These results are shown in Table 8.
>
> Comparing the results of the BART-Large policy model with and without MORL-tuning, we can tell a noticeable performance gain across most of the automatic metrics except SIM-input, **which indicates MORL effectively boosts the performance of the given policy model.**
> In Table 8, those variant MORL-BARTLarge models, namely MORL-BARTLarge w/o, produced the worst evaluation score of a specific metric among all MORL-tuned models without exception. This phenomenon reveals that removing (setting the corresponding weight to 0) a certain metric from the MORL-tuning process will forbid MORL from optimizing the policy model’s properties which can be measured by that metric. **In other words, each metric incorporated into MORL is practical to help MORL improve the policy model against the metric itself.**
>
>
>
>
> **[Significant differences in PPL]**
>
> > The analysis of the experimental results could to be further in-depth. For example, what is the reason for the significant differences in PPL between different models?
>
>
> Thanks for your advice. We strongly agree that adding a more in-depth analysis would provide more insights and enhance the persuasiveness of our research.
> Following your advice, we studied the differences in PPL, which might be attributed to the GPT model we used in the evaluation. The language model we used to calculate the PPL score is the GPT-2 model with only 137M parameters, resulting in a significant PPL difference. To check the sensitivity towards the usage of base language models, we changed it to a GPT-2 large model with 774M parameters. The results are shown below:
>
> | | PPL |
> | ----------- | ----------- |
> | Train_IFA | 32.56 |
> | Train_FA | 24.44 |
> | Dev_IFA | 33.07 |
> | Dev_FA | 31.19 |
> | Test_IFA | 35.97 |
> | Test_FA | 33.18 |
> | ControlledGen | 60.87 |
> | DeepLatentSequence | 68.45 |
> | StyleTransformer | 66.87 |
> | DeleteAndRetrieve | 34.11 |
> | SequentialTransfer | 41.19 |
> | BART-GEC | 35.83 |
> | ChatGPT | 28.84 |
> | MORL-BARTLarge | 35.65 |
> | MORL-Galactica1.3B | 34.50 |
> It is observed that the original conclusion in our manuscript still holds. We have updated these results in our revised version. Thank you for your comments!
>
>
> **[Definition of LLM]**
>
> > Some details in the paper still need to be considered. For example, although there is currently no clear definition of LLM, it is generally believed to be a decoder-only model above 1B. It seems inaccurate to include BART-Large in the category of LLM.
>
> We strongly agree with you that there is currently no clear definition for large models and it would be better to exclude BART-Large in the category of LLM. We can call BART-Large a Pre-trained Language Model (PLM). Therefore, we have revised our statement in our revised manuscript to ensure it is accurate.

---

### Official Review · Reviewer_r3ht · 2023-08-05

**Typos Grammar Style And Presentation Improvements:** 1. In Table 3, the experimental resul…
**Soundness:** 4

**Excitement:**

4: Strong: This paper deepens the understanding of some phenomenon or lowers the barriers to an existing research direction.

**Paper Topic And Main Contributions:**

This paper proposes the Academic Writing Formalization task, which is a text style transfer task and aims to rewrite informal-academic texts into formal-academic texts.
The authors construct a non-parallel training dataset for this task based on crowdsourcing, and manually annotated parallel validation and test sets.
In addition, the authors also explore the application of the metric-oriented reinforcement learning method on large language models and has achieved good results.

**Reasons To Accept:**

1. Since the previous tasks like grammatical error correction, paraphrase generation and text simplification mainly focus on the sentence level, and are not specialized and comprehensive in academic texts, the proposed task is more challenging and is of higher application value.

2. The constructed DOOLITTLE dataset is a large-scale dataset for the proposed task, which can serve as a good testbed for future research.

3. Experiments conducted on the dataset demonstrate that the MORL method can be successfully applied to large language models tuning and has achieved good results.

**Reasons To Reject:**

I have no furthur suggestions except for some typos and styles.


**Reproducibility:**

4: Could mostly reproduce the results, but there may be some variation because of sample variance or minor variations in their interpretation of the protocol or method.

**Reviewer Confidence:**

2: Willing to defend my evaluation, but it is fairly likely that I missed some details, didn't understand some central points, or can't be sure about the novelty of the work.

---

> ### Author Rebuttal · Authors · 2023-08-29
>
> Dear Reviewer r3ht,
>
> Thank you very much for your comprehensive review and valuable feedback! We have incorporated your suggestions into our revised paper including but not limited to the following edits:
>
> 1. Reorganize the methods in Table 3 and group them into three kinds. (1) the baseline models: ControlledGen, DeepLatentSequence, StyleTransformer, DeleteAndRetrieve, SequentialTransfer, and BART-GEC  (2) the proposed method: MORL-BARTLarge and MORL-Galactica1.3B   (3) ChatGPT and native rewrites. In addition, we bold the scores of MORL.
> 2. Add column borders to the table to make different metrics more distinct.
> 3. Fix typos by adding a space after parentheses at L487.
>
> Thank you again for your suggestions!

---

### Official Review · Reviewer_GKgM · 2023-08-07

**Soundness:** 4

**Excitement:**

4: Strong: This paper deepens the understanding of some phenomenon or lowers the barriers to an existing research direction.

**Paper Topic And Main Contributions:**

This study proposes the Academic Writing Formalization (AWF) task to achieve paragraph-level automatic language improvement, given the inherent difficulty of academic writing, especially for non-native English speakers. The study creates a dataset for this task, proposes a model powered with reinforcement learning, and extensively evaluates diverse baselines and their proposed models, showing their models' superiority.

**Questions For The Authors:**

- I'm just curious why the PPL scores from some (powerful) models are so high. (Table 3)
- There is a gap in the average sentence number between the training and test set. Why does this gap occur? I fear that these are intendedly controlled.
- How many different seeds are used in the experiment? Is the performance difference statistically significant?

**Reasons To Accept:**

- This is a thorough study in that it makes a series of contributions: task proposal, data generation, model proposal, and evaluation.
- The dataset is a relatively large scale, and at least from the information described in the paper, the quality seems not to be problematic.
- The proposed model shows better performance than strong baselines.

**Reasons To Reject:**

- I'm not sure about the novelty of the task characteristics. The statistics shown in Section 4.3 is informative, but I want to see the comparison with the existing language refinement benchmarks, including, e.g., JFLEG [Napoles+, 17]. I'm also curious about how the refinement is context-dependent, given that paragraph-level editing would be one important aspect of this task, and the paragraph length is somewhat short (1--2 sentences).
- Evaluation with GPT-4 was used without validating its appropriateness. At a minimum, shouldn't you show that source and target sentences have an appropriate difference in GPT-4 ratings?
- Why don't you use reference-based metrics such as GLUE? If you have some concerns about them, you should have some discussion or evaluation of the metrics on this task.

**Reproducibility:**

3: Could reproduce the results with some difficulty. The settings of parameters are underspecified or subjectively determined; the training/evaluation data are not widely available.

**Reviewer Confidence:**

3: Pretty sure, but there's a chance I missed something. Although I have a good feel for this area in general, I did not carefully check the paper's details, e.g., the math, experimental design, or novelty.

**Typos Grammar Style And Presentation Improvements:**

- Is edit-distance character-level or word-level?
- it would be better to write explicitly that the GPT-4 rates with a 5-scale value even at the evaluation. When I first looked at Table 3, I was a little confused as to what the GPT-4 score represented.

---

> ### Author Rebuttal · Authors · 2023-08-29
>
> Dear Reviewer GKgM,
>
> Thank you very much for your comprehensive review and valuable feedback! We address your comments one by one as follows:
>
> **[Comparison with other important works]**
> > I'm not sure about the novelty of the task characteristics. The statistics shown in Section 4.3 is informative, but I want to see the comparison with the existing language refinement benchmarks, including, e.g., JFLEG [Napoles+, 17]. I'm also curious about how the refinement is context-dependent, given that paragraph-level editing would be one important aspect of this task, and the paragraph length is somewhat short (1--2 sentences).
>
> Thanks for your suggestion! There are several key differences between JFLEG and our model.
>
> [Granularity] The JFLEG dataset [1] confines text refinement to the sentence level, whereas our dataset involves modifications across multiple sentences. For example, in Table 3 of JFLEG, their annotation instruction specifically mentions, "Please do not split the original sentence into two or more." In contrast, in our annotation instruction, we emphasized modifications to multiple sentences "Usually, the paragraph is fine with slight errors. For example, there are some spelling, punctuation, and capitalization problems. Redundancy problem, which means there could be a better expression, for example, split a long sentence into two short sentences."
>
> [Domain] JFLEG collected annotations based on GUG (Grammatical/Ungrammatical) corpus [2]. which contains 3.1k sentences written by English language learners for the TOEFL exam. In contrast, our data source is S2ORC and our focus lies on scientific texts, encompassing both published articles and preprints as our primary source.
>
> [Data Size] JFLEG releases a dataset of 1511 sentences with parallel edits. In contrast, our proposed dataset pushes forward with a much larger non-parallel training data and a parallel development and test dataset. There are 55.6K formal-academic paragraphs and 13.0K informal-academic paragraphs.
>
> FCE [3] and NUCLE [4] contain minimal edits because they constrain the edits by error codes.
> Lang-8 [5] is open to any internet user with a different proficiency level as Doolittle.
>
> We have included the discussion and cited these related works (e.g., [2-5]) in our revised version. Thanks!
>
> [1] Napoles, C., Sakaguchi, K., & Tetreault, J. (2017). JFLEG: A fluency corpus and benchmark for grammatical error correction.
>
> [2] Heilman, M., Cahill, A., Madnani, N., Lopez, M., Mulholland, M., & Tetreault, J. (2014, June). Predicting grammaticality on an ordinal scale.
>
> [3] Yannakoudakis, H., Briscoe, T., & Medlock, B. (2011, June). A new dataset and method for automatically grading ESOL texts.
>
> [4] Dahlmeier, D., Ng, H. T., & Wu, S. M. (2013, June). Building a large annotated corpus of learner English: The NUS corpus of learner English.
>
> [5] Tajiri, T., Komachi, M., & Matsumoto, Y. (2012, July). Tense and aspect error correction for ESL learners using global context.
>
>
> **[GPT-4 ratings]**
>
> > Evaluation with GPT-4 was used without validating its appropriateness. At a minimum, shouldn't you show that source and target sentences have an appropriate difference in GPT-4 ratings?
>
> In Table 3, we presented the GPT-4 annotation scores in terms of Academic Formality, Fluency, and Similarity of the formal-academic and informal-academic paragraphs in the DOOLITTLE test set. It is observed that there is a noticeable gap in Academic Formality and Fluency scores between those 2 sets despite their similarity. These results, along with the recent studies [6-7] in GPT-4 annotation (mentioned in Section 6.2) demonstrate the validation of GPT-4 annotation while reflecting the difficulty of our DOOLITTLE benchmark.
>
> [6] Gilardi, F., Alizadeh, M., & Kubli, M. (2023). Chatgpt outperforms crowd-workers for text-annotation tasks.
>
> [7] Zheng, L., Chiang, W. L., Sheng, Y., Zhuang, S., Wu, Z., Zhuang, Y., ... & Stoica, I. (2023). Judging LLM-as-a-judge with MT-Bench and Chatbot Arena.
>
> **[Reference-based metrics]**
>
> > Why don't you use reference-based metrics such as GLUE? If you have some concerns about them, you should have some discussion or evaluation of the metrics on this task.
>
> Thanks for your suggestions. In our original setting, we applied the semantic similarity as a reference-based metric to measure the similarity between predictions and references following [8]. Following your advice, we added more evaluation results in terms of SARI and GLEU. The results are shown below. We find that our proposed method MORL achieves significant improvements than baseline methods. Thanks for your question and we have included these new results in our revised manuscript.
>
> | | SARI | GLEU |
> | ----------- | ----------- | ----------- |
> | ControlledGen | 48.59 | 54.54 |
> | DeepLatentSequence | 37.46 | 50.4 |
> | StyleTransformer | 38.46 | 50.87 |
> | DeleteAndRetrieve | 7.98 | 1.07 |
> | SequentialTransfer | 49.17 | 71.3 |
> | BART-GEC | 69.10 | 74.72 |
> | ChatGPT | 48.84 | 70.21  |
> | MORL-BARTLarge | 55.74 | 75.75 |
> | MORL-Galactica1.3B | 63.79 | 78.37 |
>
>
> [8] Wieting, J., Berg-Kirkpatrick, T., Gimpel, K., & Neubig, G. (2019). Beyond BLEU: training neural machine translation with semantic similarity.
>
>
> **[PPL]**
>
> > I'm just curious why the PPL scores from some (powerful) models are so high. (Table 3)
>
> The language model we used to calculate the PPL score is the GPT-2 model with only 137M parameters, which is not that powerful. To check the sensitivity towards the usage of base language models, we changed it to a GPT-2 large model with 774M parameters. The results are shown below:
> | | PPL |
> | ----------- | ----------- |
> | Train_IFA | 32.56 |
> | Train_FA | 24.44 |
> | Dev_IFA | 33.07 |
> | Dev_FA | 31.19 |
> | Test_IFA | 35.97 |
> | Test_FA | 33.18 |
> | ControlledGen | 60.87 |
> | DeepLatentSequence | 68.45 |
> | StyleTransformer | 66.87 |
> | DeleteAndRetrieve | 34.11 |
> | SequentialTransfer | 41.19 |
> | BART-GEC | 35.83 |
> | ChatGPT | 28.84 |
> | MORL-BARTLarge | 35.65 |
> | MORL-Galactica1.3B | 34.50 |
>
> It is observed that the original conclusion in our manuscript still holds. We have updated these results in our revised version. Thank you for your comments!
>
> **[Paragraph length]**
>
> > I'm also curious about how the refinement is context-dependent, given that paragraph-level editing would be one important aspect of this task, and the paragraph length is somewhat short (1--2 sentences). There is a gap in the average sentence number between the training and test set. Why does this gap occur? I fear that these are intendedly controlled.
>
> We reviewed Table 2 and noticed a few minor discrepancies regarding the statistics of "paragraph length”. Please find the up-to-date statistics below (This is exactly the dataset used in all the experiments). Sorry for the typo. Thank you for pointing this out and we have updated our manuscript with the accurate statistics.
>
> | | | P# | S# | V# | Avg. Words | Avg. Sent. |
> | ----------- | ----------- | ----------- | ----------- | ----------- | ----------- | ----------- |
> | Train | FA | 55.6K | 172.8K | 84.3K | 51.42 | 3.11 |
> | Train | IFA | 13.0K | 41.3K | 38.9K | 52.17 | 3.17 |
> | Dev | FA | 465 | 1359 | 5.2K | 47.33 | 2.92 |
> | Dev | IFA | 465 | 1362 | 5.3K | 47.79 | 2.92 |
> | Test | FA | 415 | 927 | 4.4K | 42.52 | 2.23 |
> | Test | IFA | 415 | 910 | 4.5K | 43.08 | 2.19 |
>
> As for the paragraph length, we can see that it is 2-3 sentences and paragraph level edits could be done. Especially, the average words are rather accurate and we can see that the paragraph is quite long and some paragraphs consist of long sentences separated with multiple commas.
>
> As for the gap between the training and test set in terms of average sentence number, it is because annotators will filter out the data that can not meet the required standards.
>
> **[Seeds]**
> > How many different seeds are used in the experiment? Is the performance difference statistically significant?
>
> We ran our experiments three times and the improvements of MORL over all baselines are statistically significant under the t-test with p < 0.05. We will report this significance in our revised paper.
>
> **[Edit-distance]**
> > Is edit-distance character-level or word-level?
>
> The edit-distance reported in our paper is character-level. Thanks for your comments and we have added this detail in our revised version.
>
> **[GPT-4 with a 5-scale value]**
> > it would be better to write explicitly that the GPT-4 rates with a 5-scale value even at the evaluation. When I first looked at Table 3, I was a little confused as to what the GPT-4 score represented.
>
> Thank you for your suggestion! We have clarified that GPT-4 rates with a 5-scale value at the evaluation in our revised paper following your advice.

---

### Official Review · Reviewer_euze · 2023-08-10

**Soundness:** 4

**Excitement:**

3: Ambivalent: It has merits (e.g., it reports state-of-the-art results, the idea is nice), but there are key weaknesses (e.g., it describes incremental work), and it can significantly benefit from another round of revision. However, I won't object to accepting it if my co-reviewers champion it.

**Paper Topic And Main Contributions:**

This paper focuses on a study of AWF tasks to embellish academic language, which helps to improve the quality of essays in formal writing. The paper introduces the AWF task to break through the limitations of traditional language touch-up methods. In order to solve the problem of formal text conversion, the paper also introduces the MORL method, which, by combining reinforcement learning techniques and metric optimization, enables the language model to be trained with varying degrees of automated feedback, thereby improving the quality of the generated formal academic text to address the challenge of academic writing quality improvement, and at the same time facilitates the automated generation and improvement of formal academic text.

**Reasons To Accept:**

The paper applies MORL to LLM, using essentially real formal academic texts from DOOLITTLE as a dataset and incorporating varying degrees of automated feedback in the training process. The paper's dataset is of high quality and the training method is relatively new, while the paper is more compatible with this journal.

**Reasons To Reject:**

This study is for mentioning practical application scenarios, i.e., it does not mention how these methods can be applied to real scenarios in order to solve real problems. The experimental results were compared with other large models such as chat-GPT, as well as with essentially real formal academic texts in DOOLITTLE, but no comparisons were made between academic embellishments and manual academic embellishments on a particular text, so it is not clear what their practical value is. At the same time, academic writing involves multiple disciplines, although there are academic texts from multiple disciplines in the database, it does not mean that there is no difference in the level of academic embellishment in various disciplines, and a comparison between various disciplines can be considered in the future.

**Reproducibility:**

4: Could mostly reproduce the results, but there may be some variation because of sample variance or minor variations in their interpretation of the protocol or method.

**Reviewer Confidence:**

4: Quite sure. I tried to check the important points carefully. It's unlikely, though conceivable, that I missed something that should affect my ratings.

---

> ### Author Rebuttal · Authors · 2023-08-29
>
> Dear Reviewer euze,
>
> Thank you very much for your comprehensive review and valuable feedback! We address your comments one by one as follows:
>
> **[Application scenarios]**
>
> > This study is for mentioning practical application scenarios, i.e., it does not mention how these methods can be applied to real scenarios in order to solve real problems. The experimental results were compared with other large models such as chat-GPT, as well as with essentially real formal academic texts in DOOLITTLE, but no comparisons were made between academic embellishments and manual academic embellishments on a particular text, so it is not clear what their practical value is.
>
> Thanks for your comments! Our proposed task and dataset is an advancement for grammatical error correction (GEC). We hope our task could be applied to improve academic writing, given the inherent difficulty of academic writing, especially for non-native English speakers. In Section 3: Dataset Construction, we put a lot of effort into making our data more practical and aligned with academic writing, including data collection, quality control, and especially for our dev/test set, where all ground truth samples were manually annotated. Thus, we believe that our benchmark can well represent real-world scenarios and the experimental results on DOOLITTLE can reflect the practical values of all baselines. For practical application scenarios, our dataset can be used to train a model for automatically polishing academic writing, similar to Grammarly.
>
> **[Disciplines]**
>
> > At the same time, academic writing involves multiple disciplines, although there are academic texts from multiple disciplines in the database, it does not mean that there is no difference in the level of academic embellishment in various disciplines, and a comparison between various disciplines can be considered in the future.
>
> Thanks for your advice! We consider the problem to be a trade-off between the universality and specialty of our dataset. In the current stage, we propose DOOLITTLE as a general Academic Writing Formalization benchmark where the utilization of a broader range of disciplines is included to highlight its comprehensiveness. We believe that studying the difference in the level of academic embellishment across various disciplines through comparison must be a promising direction for further study. We’ll explore this in future work.
> We have updated our manuscript with these details and discussions. Thanks very much for your constructive suggestion!

---

### Meta-Review · Area_Chair_ayG2 · 2023-09-19

**Recommendation:** 5

**Metareview:**

The paper has received largely favorable reviews from all the reviewers. Some concerns which were raised in the earlier reviews have been adequately addressed as summarised below

1. High quality datasets [concerns about differences from existing datasets have been adequately addressed]
2. Evaluation using metrics in addition to GPT 4 [results using SARI and GLUE have been added during the response period]
3. Usefulness of the MORL method (ablation results have been provided to show that the proposed method MORL indeed helps]

Overall this is a good paper which contributes a dataset, a new method and does a thorough comparison with appropriate ablation studies.

I request the authors to include the new results provided during the response period in to the main body or appendix of the paper.

---

### Decision · Program_Chairs · 2023-10-07

**Decision:**

Accept-Main

**Comment:**

The paper has received largely favorable reviews from all the reviewers. Some concerns which were raised in the earlier reviews have been adequately addressed as summarised below

1. High quality datasets [concerns about differences from existing datasets have been adequately addressed]
2. Evaluation using metrics in addition to GPT 4 [results using SARI and GLUE have been added during the response period]
3. Usefulness of the MORL method (ablation results have been provided to show that the proposed method MORL indeed helps]

Overall this is a good paper which contributes a dataset, a new method and does a thorough comparison with appropriate ablation studies.

I request the authors to include the new results provided during the response period in to the main body or appendix of the paper.